# Implication for alphavirus host-cell entry and assembly indicated by a 3.5Å resolution cryo-EM structure

Lihong Chen[1,2,3], Ming Wang[2], Dongjie Zhu [1,4], Zhenzhao Sun[2], Jun Ma[1], Jinglin Wang[5], Lingfei Kong[1], Shida Wang[2], Zaisi Liu[2], Lili Wei[2], Yuwen He[5], Jingfei Wang[2] & Xinzheng Zhang[1,3]

Alphaviruses are enveloped RNA viruses that contain several human pathogens. Due to intrinsic heterogeneity of alphavirus particles, a high resolution structure of the virion is currently lacking. Here we provide a 3.5 Å cryo-EM structure of Sindbis virus, using block based reconstruction method that overcomes the heterogeneity problem. Our structural analysis identifies a number of conserved residues that play pivotal roles in the virus life cycle. We identify a hydrophobic pocket in the subdomain D of E2 protein that is stabilized by an unknown pocket factor near the viral membrane. Residues in the pocket are conserved in different alphaviruses. The pocket strengthens the interactions of the E1/E2 heterodimer and may facilitate virus assembly. Our study provides structural insights into alphaviruses that may inform the design of drugs and vaccines.

[1] National Laboratory of Biomacromolecules, CAS Center for Excellence in Biomacromolecules, Institute of Biophysics, Chinese Academy of Sciences, 100101 Beijing, People's Republic of China. [2] State Key Laboratory of Veterinary Biotechnology, Harbin Veterinary Research Institute, Chinese Academy of Agricultural Sciences, 150069 Harbin, People's Republic of China. [3] University of Chinese Academy of Sciences, 100049 Beijing, People's Republic of China. [4] School of Life Science, University of Science and Technology of China, 230026 Hefei, People's Republic of China. [5] Yunnan Tropical and Subtropical Animal Viral Disease Laboratory, Yunnan Animal Science and Veterinary Institute, Kunming 650224, People's Republic of China. These authors contributed equally: Lihong Chen, Ming Wang, Dongjie Zhu, Zhenzhao Sun. Correspondence and requests for materials should be addressed to J.W. (email: wangjingfei@caas.cn) or to X.Z. (email: xzzhang@ibp.ac.cn)

Alphaviruses are enveloped, single-stranded RNA viruses with a diameter of ~700 Å. A considerable number of these viruses pose as lethal human pathogens[1], including Sindbis (SINV), Semliki Forest (SFV), Ross River, Chikungunya (CHIV) and Venezuelan equine encephalitis (VEEV) virus. Currently, there are neither any licensed vaccines nor anti-viral therapeutics available for treating infections caused by alphaviruses.

The $T = 4$ icosahedral symmetry alphaviruses comprises 80 trimer, including 60 quasi-three-fold symmetry trimers as well as 20 icosahedral three-fold symmetry trimers. In addition, these viruses possess a nucleocapsid core with 240 copies of capsid protein (CP).

Each trimer is composed of three E1/E2 heterodimers. Both E1 and E2 are transmembrane proteins that are anchored to the viral membrane with their transmembrane helices associated with each other. The E2 protein is responsible for host cell receptor recognition. It contains A, B, and C ectodomains as well as one subdomain D, which is composed of a loop and a helix located near the viral membrane connecting the Domain C with the transmembrane helix (TM)[2]. E1 protein, a homolog to the Flavivirus E glycoprotein, takes responsibility for membrane fusion. E1 possesses ectodomains I, II, III (namely DI, DII and DIII) with the fusion loop located at the distal domain of DII and a stem region that connects DIII with the E1 TM. The stem region forms a loop in the virus[2] and packs against the homotrimer core, at least as observed in the trimeric and fusogenic crystal structure of E1[3].

Alphaviruses enter the host cells via a receptor-mediated endocytosis process[4]. After endocytosis, the acidic environment of the endosome causes an irreversible conformational rearrangement of E1 and E2 envelope glycoproteins in which E2 dissociates from E1. This is followed by E1 forming homotrimers exposing hydrophobic fusion loops. The stem loop of the virus becomes exposed upon dissociation of E1/E2 heterodimer. The stem region subsequently packs against the trimer core at a relatively late step in the formation of the final post-fusion hairpin, after the fold-back of DIII[5]. The formation of the homotrimers triggers the fusion of the viral membrane with the endosomal membrane, which results in the release of viral genome into the cytoplasm.

During the assembly of virus particles, the CP is autoproteolytically processed, thus forming the nascent nucleocapsid core (NC). The trimers formed from virus-encoded E1-E2 envelope glycoproteins heterodimers are transported to the plasma membrane. They interact with the nascent NCs in the cytoplasm to form intact progeny viruses which can bud out of the host cell for a new round of infection. The cytoplasmic tail of E2 binds to a hydrophobic pocket in the C-terminal domain of the CP, which links the protein of the inner core with those on the surface of the virion. Interestingly, CP mutants that fail to preassemble cytoplasmic NCs still form viral particles with the same $T = 4$ icosahedral symmetry, albeit with much lower efficiency[6]. This phenomenon suggests that particle budding is promoted both by preassembled NCs and lateral interactions of envelope proteins[7]. The stem loop of E1 is important for E1/E2 dimerization and virus assembly; however, no specific stem sequence is required for membrane fusion[8]. Mutagenesis studies of the E2 subdomain D loop have revealed that several of the conserved histidines from these regions reduce the thermo-stability of the virus and reduce viral growth from 1.6 to 2 log due to inhibition of a late stage of the assembly of the virus at the plasma membrane[9]. Additionally, mutations in TMs affect heterodimer stability and result in budding defects[10].

The ectodomain structures of the E1–E2 heterodimers of different alphaviruses under neutral pH or a low pH conditions have been studied by X-ray crystallography[11–15]. In addition, the complete structures of different alphaviruses have been extensively studied using cryo-electron microscopy[2,16–22]. However,

the resolution of these structures was lower compared with those reported for other icosahedral viruses of similar size. This could probably be due to the lack of homogeneity of alphaviruses. To date, the highest resolution reported for an intact virion is a 4.8 Å resolution map of VEEV[2]. While with the help of crystal structures, these cryo-EM studies have provided important insights into the folding of full-length structure of E1, E2, CP, and E3 and the interactions among them. However, the lack of side-chain densities in these cryo-EM structures failed to elucidate the precise interactions that between E1 and E2 proteins near the subdomain D of E2 protein and between the E2 proteins in neighboring trimers. A more precise understanding of the mechanism of assembly and host-cell entry of alphaviruses requires a higher resolution structure of the viruses.

Here, we obtained a 3.5 Å resolution map of SINV using a "block-based" reconstruction method[23] we recently developed. Our structure would enable analysis of side chain-side chain interactions on the whole glycoproteins. It reveals several new structural features and sheds light on the detailed assembly and host-cell entry mechanisms of alphaviruses.

## Results

**Overall structure of SINV.** The structure of SINV (Supplementary Figure 1A and B) was first determined to ~4.3 Å using the JSPR[24] software package assuming an icosahedral symmetry for the reconstruction. In order to overcome the heterogeneity of the sample and to further push the resolution, we used block-based reconstruction method[23] to further refine the structure. In the block-based reconstruction method[23], three blocks with one block containing five trimers located near icosahedral five-fold symmetry axis, another block containing four trimers located near icosahedral three-fold symmetry axis and the third block containing 17 CPs were selected (Fig. 1a). Each block was refined and reconstructed separately (see Methods). The structure of the entire virus was calculated by combining the densities of the three blocks to give rise to the asymmetric unit followed by imposition of the icosahedral symmetry. The resolutions of the envelope glycoprotein and capsid core were determined to 3.5 and 4.7 Å by gold standard Fourier Shell Correlations (FSC), respectively (Supplementary Figure 1C). As shown in Supplementary Figure 1D, the local resolution of the map (see Methods) confirms the resolution assignment. The atomic model of the glycoproteins was built ab-initially (Supplementary Figure 1E). This model includes proteins E1, E2, and E3 (Fig. 2a). The model of capsid core was built by fitting the structure of the capsid (PDB code1EP5) into the density.

Comparison of our structure of E1–E2 ectodomains with SINV E1–E2 crystal structure determined at low pH[11], CHIKV E1–E2 crystal structure[15] and cryo-EM VEEV E1–E2 ectodomains[2] showed that the overall structure of these domains is similar (Supplementary Table 1).

Our analysis showed that in each E1–E2–E3 molecule, four N-linked glycosylation sites are present, namely N139, N245 on E1, N283 on E2 and N14 on E3 (Supplementary Figure 2). We found three glycosylation sites on E1 and E2 which is in agreement with results of previous studies[11,17]. However, we were unable to confirm the other previously identified glycosylation site on E2 N191, as this region of domain B was disordered in our structure.

**β-ribbon connector and its interactions with E3 and Domain B.** The β-ribbon connector is important for the conformational changes of alphaviruses triggered by low pH conditions. An earlier study[11] found this connector is partially disordered in the low pH E1–E2 crystal structure. However, in our experiment condition (pH 8.0), this connector in our

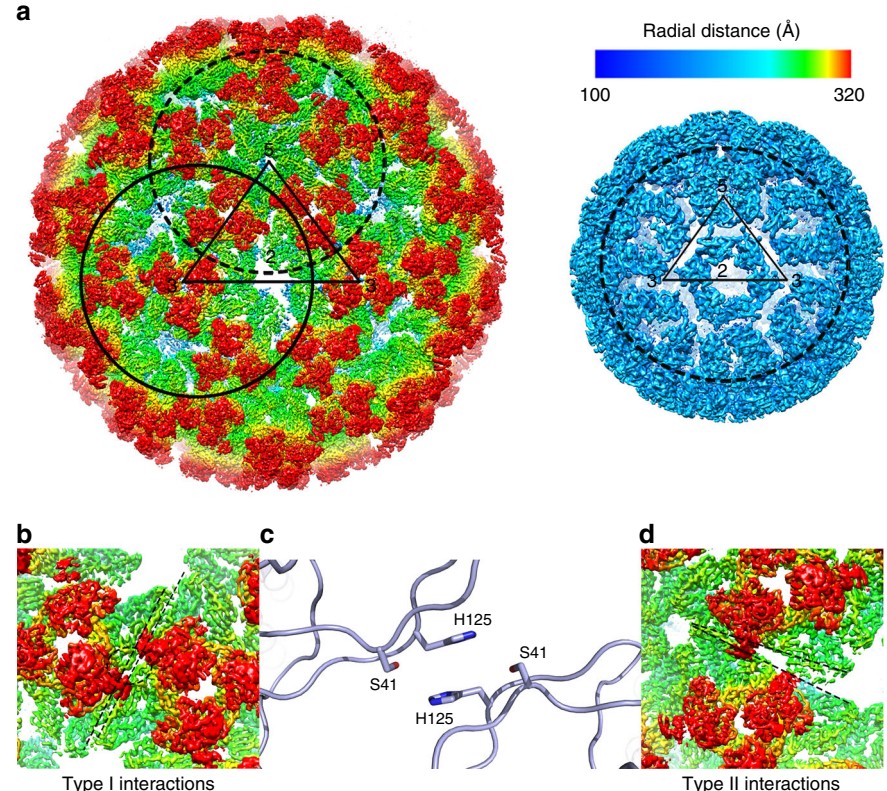

**Fig. 1** Overall structure of SINV and the two types of inter-triangle interactions. **a** Radially colored 3D reconstruction of SINV. Left: an asymmetric unit on the cryo-EM density map of SINV is enclosed by a black dash triangle. Block 1 and block 2 containing five trimers near the icosahedral five-fold axis and four trimers near the icosahedral three-fold axis are indicated by dash black cycle and solid black cycle, respectively. Right: block 3 is marked by dash black cycle on the inner capsid shell. **b** The type I interactions between a quasi-three-fold trimer and an icosahedral three-fold trimer are enlarged. Two parallel dash lines indicate the interacted edges of two neighboring trimers. **c** The third region of the contacts is shown. The H125 forms hydrogen bond with S41 on the other side of the edge. **d** The type II interactions between two quasi-three-fold trimers are enlarged. The distance between two edges becomes larger when away from the five-fold axis. Two dash lines indicate the interacted edges of two neighboring trimers. The maps in **b**, **d** are colored according to the radial scale in **a**

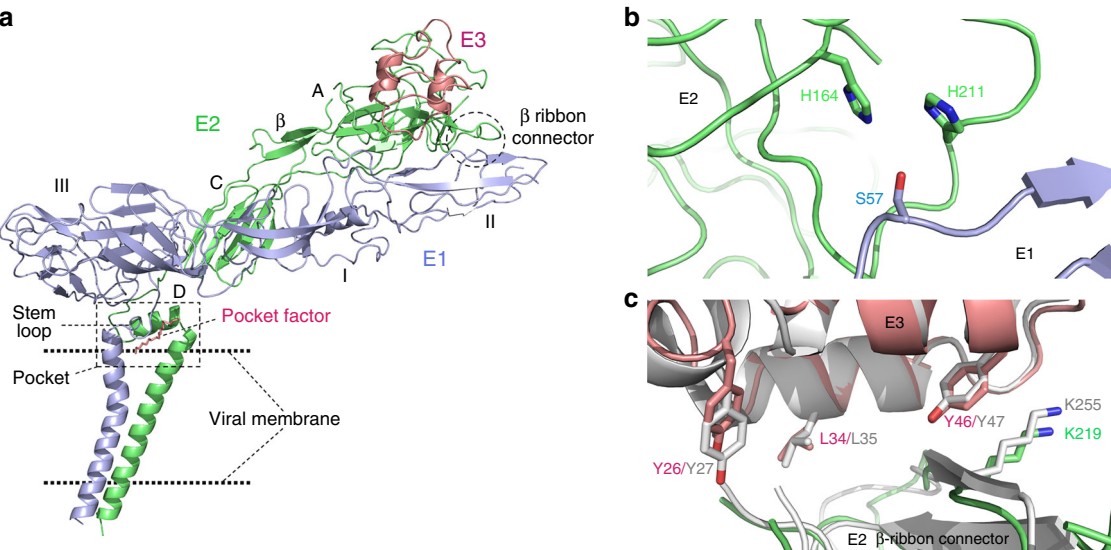

**Fig. 2** The E1/E2/E3 molecules and the pivotal interactions near β-ribbon connector. **a** Atomic model of E1/E2/E3 was built ab-initially, which includes domain I, II and III of E1, domain A, B, C and subdomain D of E2, stem region of E1, and TMs of E1 and E2. **b** E2 H164 on β ribbon connector (colored by green) forms hydrogen bond with E1 S57 (colored in purple). E2 H211 is also involved into this interaction. **c** The interactions between E3 (colored by pink) and β-ribbon connector (colored in green) are shown with the side chain of the complete conserved amino acids that is involved in the interactions being displayed. The crystal structure of CHIKV E1-E2-E3 (PDB code 3N44) (colored in gray) are superimposed with SINV based on the structural similarity of E3

structure is ordered and is exhibiting a similar structure to that observed in structures of alphaviruses as well as in x-ray crystal structures obtained at neutral pH conditions[2,15].

In addition to the connector structure, we confirmed the presence of a hydrogen bond between the side chain of E2 H170 and E1 S57 (corresponding to SINV E2 H164 and E1 S57), as shown previously in the CHIKV E1-E2 crystal structure[15]. This bond appears to be preserved in all mosquito-borne alphaviruses. However, this interaction was not found in cryo-EM CHIKV structure[22]. Moreover, as shown in Fig. 2b, E2 H211 also participates in this interaction by forming an additional hydrogen bond with E2 H164 through their side chain nitrogens. At low pH, the interaction between E2 H164 and H211 becomes repulsive, which may assist in disrupting the interaction between E2 H164 and E1 S57. However, E2 H211 is not completely conserved among alphaviruses, suggesting that it may cause different responses of different alphaviruses to the pH change at early stage of viral entry into host cells[25].

Compared with the surrounding densities, the domain B exhibits much weaker and more disordered densities. In the same region, we found that 12 amino acids of E1 protein including the E1 fusion loop are also disordered. In comparison, these regions are more ordered in the VEEV structure as reported previously[2]. Primary sequence alignment studies reveal that the domain B of this SINV strain lacks 30 amino acids when compared with the sequences of other alphaviruses. A lack of these amino acids probably reduces the number of interactions between the β-ribbon connector and domain B, resulting in a more disordered domain B. The fusion loop has been found to be associated with domain B[2,15]. Such association may cause disordered densities of the fusion loop in our structure.

As shown in Fig. 2c, the cleaved E3 domain of our structure is associated with the β-ribbon connector, which differs from previously described cryo-EM structures of SINV[11], showing that cleaved E3 is released. However, such an association was observed in other alphaviruses[2,26]. It was found that in SFV, the cleaved E3 is gradually released from virus at 37 °C and under neutral pH conditions with a halftime of 0.5 h[27]. These observations indicate that purification conditions and the time point of the sample freezing greatly affect the structure obtained.

Next, we assessed the interaction between E3 and the β-ribbon connector. As shown in Fig. 2c, the E3 domain interacts with the β-ribbon connector mostly via hydrophobic interactions, which is in agreement with a previously reported structure of CHIKV E1–E2–E3[15]. These hydrophobic interactions are stable in the acidic pH. Therefore the uncleaved or cleaved E3 can stabilize and locate the β-ribbon connector through these interactions when E1–E2 heterodimers are processed through the Golgi and trans-Golgi network. On the other hand, these hydrophobic interactions are not strong, allowing E3 to be partially released at neutral pH after the virus is released from host cells, thus to primes the conformational change of the trimer for low-pH activation. Most of the residues involved in the hydrophobic interactions, such as L34, Y46 and Y26 of the E3 and K219 of the β-ribbon connector, are conserved in all mosquito-borne alphaviruses (Supplementary Figure 3). Together, these observations indicate that the interactions (Fig. 2c) between E3 and the β-ribbon connector of alphaviruses are highly conserved. Importantly, these interactions have been endowed with a similar strength over the course of the evolution, conferring a conserved function on E3[27].

**Inter-triangle interactions**. The 80 trimers assemble into an icosahedral lattice through trimer–trimer interactions.

In order to evaluate the amino acids that are involved in the trimer-trimer interactions, we analyzed these regions in our

structure. Two types of trimer-trimer interactions, the interactions between a quasi-three-fold trimer and an icosahedral three-fold trimer (type I) and the interactions between two neighboring quasi-three-fold trimers (type II), are observed in this structure.

As shown in Fig. 1b, in the type I interactions, the two trimers have a 150 Å-long boundary where we were able to find five regions of contacts between the two trimers. The interactions exhibit a quasi-two-fold symmetry with the third region located at the quasi-two-fold axis. This region is mainly formed by the interactions between two completely conserved histidines (E1 H125) from the adjacent trimers (Fig. 1c). Such an interaction has been suggested[3] previously based on the results from fitting the X-ray structure of SFV E1 into a low resolution cryo-EM map of SFV. The conformational change required for the formation of the fusogenic E1 trimer during virus entry may start from the break of this interaction, which turn into a repulsive force once the pH drops below the pKa of the histidine.

In contrast, the type II interactions do not follow the quasi-two-fold symmetry (Fig. 1d). Throughout the 150 Å-long boundary formed between neighboring trimers, only one region forms contacts between the two E1 molecules. The two E1 molecules are separated further apart away from the five-fold axis, which is required to maintain the curvature of the virus. The relatively weaker type II interactions are located near the icosahedral five-fold axis, suggesting that during the virus entry, the trimers near the five-fold axis may be easy to disassembly and to form fusogenic E1 trimers.

The contacts of the type I and II interactions between the trimers are shown in the Supplementary Table 2. The amino acids that are involved into the identified inter-triangle contacts are either completely or highly conserved. The 80 trimers are organized into an icosahedral symmetry virus through 60 type I and type II interactions. These interactions are relatively weak when compared with other icosahedral viruses of similar size[28–30], which could possibly lead to an inaccurate organization of the trimers and cause certain flexibilities in the glycoprotein shell of the virus.

**Hydrophobic pocket near the viral membrane**. The 3.5 Å resolution map of SINV enabled us to build an accurate model for E2 subdomain D, E1 and E2 transmembrane (TM) helix as well as the E1 stem region (Fig. 3a). Our structure confirms the previous observation in the structure of VEEV that the E1 TM helix is formed by two consecutive helices separated by a kink region where the conserved GG motif is facing away from the E2 TM helix[2].

Above the kink region, a hydrophobic pocket is formed by E2 subdomain D, E1 and E2 TM helices (Fig. 3a, d). Several complete conserved and highly conserved amino acids were found to form this pocket (Fig. 3b). A 20 Å long linear molecule parallel to the viral membrane (Fig. 3a, c) is sitting inside this pocket. The molecule stabilizes the pocket by forming several hydrophobic interactions with the surrounding proteins (Table 1), among which, the interactions involve amino acids W409 and Y324, which are conserved in all mosquito-borne alphaviruses. In addition to the pocket factor, the hydrophobic pocket is stabilized by several hydrogen bonds connecting E2 subdomain D and E1 near the viral membrane. The interactions include several hydrogen bonds centered on a crucial hydrogen bond between the side chains of E2 H314 of the subdomain D loop and E1 S403 of the E1 TM. In addition, E1 W 409 belonging to the E1 stem region forms hydrophobic interaction with another completely conserved amino acid E2 P317 on E2 subdomain (Fig. 3c). A candidate of the pocket factor is the hydrophobic phospholipid tail, which is fitted well into the linear density (Fig. 3c). Such a pocket factor filled pocket is common in alphaviruses. A global

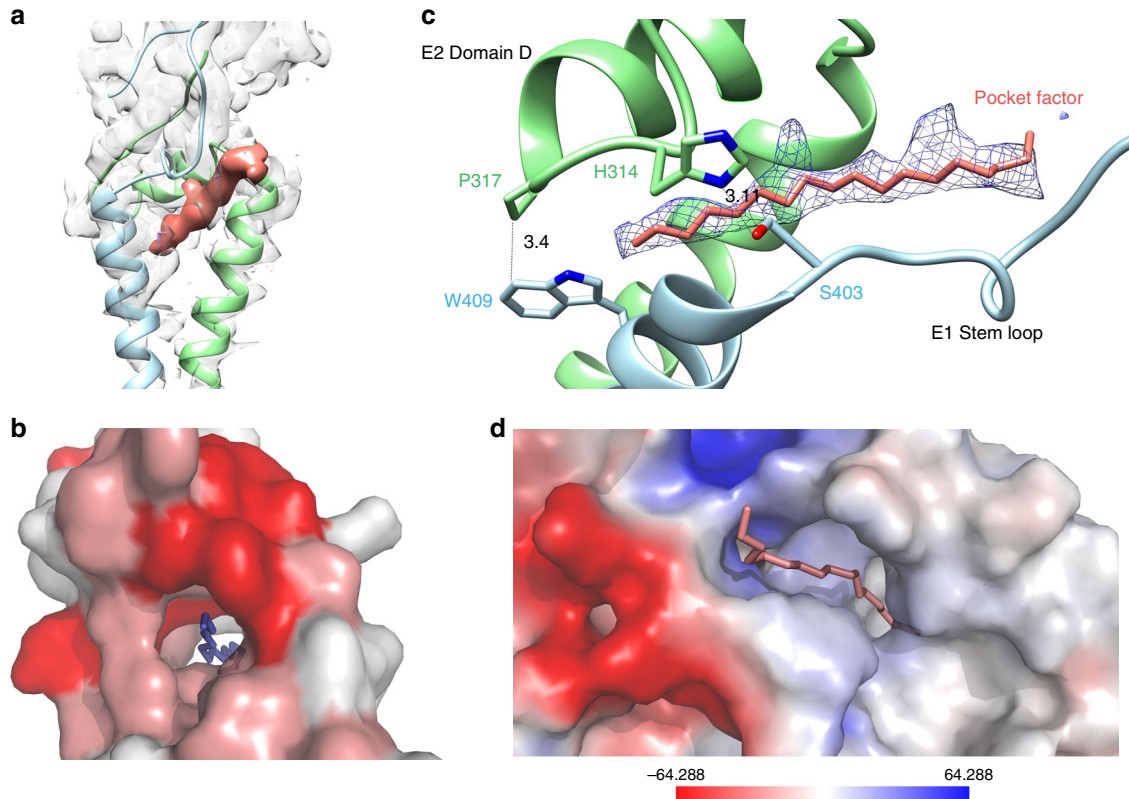

**Fig. 3** hydrophobic pocket and the pocket factor. **a** A pocket is formed by E2 (green) subdomain D, E2 TM helix and E1 (blue) TM helix, depicted in ribbons, a 20 Å-long pocket factor was found to fill the pocket. The cryo-EM density of the pocket factor is colored in brown. The cryo-EM density of E1 and E2 are shown in gray with transparency value of 0.5. **b** The completely conserved (deep red) and highly conserved (pale red) amino acids composing the pocket are shown. **c** Hydrophobic phospholipid tail can be fitted well into the density of the pocket factor. The two completely conserved amino acids E2 P317 and E1 W409 forming hydrophobic interaction are shown. On the other side, E2 H314 forms hydrogen bonding with E1 S403. **d** It is hydrophobic inside the pocket. The surface of the protein is colored by surface potential

| Table 1 Amino acids in the pocket that form hydrogen bonds with the pocket factor | |
| --- | --- |
| **E1** | **E2** |
| F398 | I320[a] |
| W409[a] | Y324[a] |
| L410[b] | I335[b] |

[a]Completely conserved in mosquito-borne alphaviruses
[b]Highly conserved in mosquito-borne alphaviruses

density has been observed in VEEV in a location similar to that of this pocket factor[2]. Considering the fact that the VEEV map is an average of four E1/E2 molecules in one asymmetric unit and is at relative lower resolution, it is possible that this global density represents the pocket factor identified in our structure.

Below the kink region, a rigid helix bundle formed via hydrophobic interactions between the E1 and E2 TM helices together with the E2 cytoplasmic tail of the C-terminal of the E2 TM helix helps locate the hydrophobic pore of NP in the nascent icosahedral nucleocapsid core.

Similar to VEEV[2], SINV contains four histidines in its E2 subdomain D. The protonation of these histidines in a low pH environment could strengthen the interaction between the subdomain and the neighboring negatively charged lipid head groups. Together with the disruption of the contact between E2 H314 and E1 S403 at low pH, the changes resulting from exposure to acidic environment may cause the collapse of the

pocket and the release of pocket factor. A previous study of the cryo-EM structure of SFV at pH of 5.9 showed a conformational change of transmembrane region that forms this pocket[31]. Together, these indicate the disassembly of this hydrophobic pocket during virus entry which results in a decrease in the interaction between E1 and E2 and assists in the formation of the fusogenic E1 trimer.

**Stem loop**. As shown in Supplementary Figure 3 and 4A, the N-terminal of the stem loop from E1 P383 to V388 is associated with E2 domain C via hydrogen bonding interaction between the completely conserved residue E1 H386 and the highly conserved E2 S308. In addition to this interaction, the backbone amide and carbonyl groups on E1 V388 are hydrogen bonded with main chain O and N on E1 W304, respectively. Such interactions help prevent the exposure of the stem loop on the assembled virus.

Furthermore, the region encompassing amino acids E1 N389 to N394 of the stem loop bridges E1 DIII with E2 C domain via formation of several hydrogen bonds involving E1 H392 and N394 (Supplementary Figure 3 and Fig. 4a). These hydrogen bonds strengthen the interaction between E1 and E2, which is important for assembly of the virus. At acidic pH, both the interactions formed by E1 H386 and H392 become weaker when compared to those at neutral pH. The loss of interaction containing E1 H392 facilitates the dissociation of E1 and E2 during virus entry. The weakening of both interactions helps to free the stem loop and the free stem loop packs against the trimer core to form post-fusion hairpin at a relatively late stage of the fusion process.

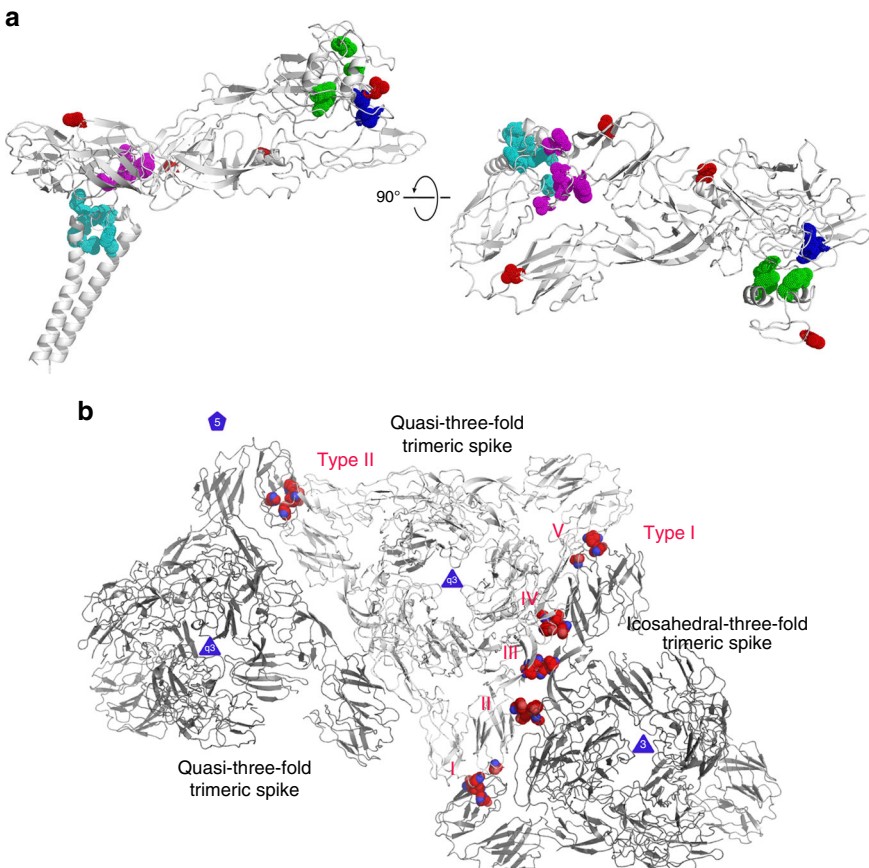

**Fig. 4** Amino acids (displayed as dot in Pymol) that involved in the key interactions which identified or discussed in this SINV structure. **a** The completely or highly conserved amino acids of E1 and E2 that are involved in the interactions of forming the hydrophobic pocket and forming the hydrophobic contact with pocket factor are colored in cyan. The amino acids of E1 that are involved in connecting the stem loop with E1 DIII and E2 domain C are colored in magenta. The completely conserved amino acids of E3 and E2 that are involved in hydrophobic interaction between E3 and β-ribbon connector are colored in green. The completely or highly conserved amino acids of E1 S57, E2 H164 and H211 that form an interaction at neutral pH but become repulsive at low pH to facilitate the dissociation of β-ribbon connector from E1 are colored in blue. The identified glycosylation sites are colored in red. **b** Amino acids of E1 that are involved in the type I and II inter-triangle interactions are colored in red

## Discussion

The loose interactions between trimers suggest that the efficiency of the assembly of the trimers into an icosahedral lattice by the trimer itself is low, which has been already observed in the assembly of alphavirus harboring CP mutants[6]. However, the loose interactions between spikes could facilitate the reorganization of E1 and E2 during pH induced conformational changes.

The ectodomain of the trimer and the hydrophobic pocket of NP form rigid connection through the helix bundle followed by the stable hydrophobic pocket. Thus, the position of the ectodomain of the trimer outside the viral membrane can be determined by the hydrophobic pocket of NP inside the viral membrane. Taking into account the weak interactions between the neighboring trimers, the accurate location of the trimers outside the viral membrane might help initialize the right interactions between trimers and benefit the assembly of the trimers into an icosahedral lattice.

Based on the structure of VEEV, it has been suggested[2] that conserved contacts might form between E2 H348 and E1 S403 and between E2 H352 and E1 W409. Our results supports the previous proposal that the SFV E2 H348 (corresponding to SINV E2 H314) contacts E1 S403 (corresponding to SINV E1 S403). However, E2 H318 (SFV E2 H352) fails to contact E1 W409 (SFV E1 W409). Previous studies on SFV[9] showed that although the

mutation of E2 H348/H352 causes an unexpected budding defect, it failed to affect the fusion of the E1 subunits. Our structure indicates that the loss of the contact of E2 H314 with E1 S403 may result in the collapse of the pocket. It may cause the trimer outside the viral membrane to be located inaccurately, thus hindering the interaction between trimer and the formation of the glycoprotein lattice. On the other hand, the contact of E2 H314 with E1 S403 will be lost in an acidic environment during the virus entry before the formation of the fusogenic E1 trimer. Therefore, the mutation of SFV E2 H 348 does not affect the fusion.

In our structure, several completely conserved histidines were identified to be involved either in the formation of inter-triangle-contacts (Fig. 4b, Supplementary Figure 4) or for stabilizing the E1/E2 heterodimer (Fig. 4a, Supplementary Figure 4). These histidines play an important role in the subsequent conformational change of the virus at acidic pH during the entry of the virus into host cells, which results in the disruption of inter-triangle-contacts and the dissociation of the heterodimer. Moreover, in the region of the pocket and the TMs bundle, the E1/E2 heterodimer is stabilized through hydrophobic interactions which are less sensitive to pH compared to the hydrophilic interactions. Therefore, the complete dissociation of E1 from E2 may require a pH lower than that needed for the dissociation of β-ribbon connector from E1[11,15,25].

A hydrophobic pocket factor has previously been shown to stabilize a number of Picornaviruses during assembly. In contrast, the expulsion of this factor after receptor recognition destabilizes the viral capsid, thus facilitating the release of the genome during infection. Here, we show that alphaviruses possess a similar pocket factor. Comparable to the factor in Picornaviruses, it stabilizes the assembly of the alphaviruses, and may be released under acidic pH environmental conditions, aiding the formation of the fusogenic trimer for the release of the viral genome. However, the detailed chemical composition of the pocket factor and the molecular mechanisms involved in its release remain to be elucidated. Small molecules that bind the Picornaviruses pocket tighter than native pocket factor have been designed as virus inhibitors[32–34]. It is possible that a similar tight-binding molecule could be a drag candidate for alphaviruses.

## Methods

**Virus production and purification.** SINV strain YN_222 (MH229928) was grown in Vero cells (ATCC, CCL-81) at a multiplicity of infection (MOI) of 0.1. Virus was harvested 24 h post infection and centrifuged to remove cell debris. The supernatant containing the virus was concentrated by pelleting through a 20% (w/v) sucrose cushion at $75,000 \times g$ for 2 h in a Beckman rotor (32 Ti) at 4 °C. The virus pellet was gently resuspended in 2 ml PBS buffer and loaded onto a 25–45% (w/v) sucrose discontinuous density gradient. This gradient was centrifuged at $75,000 \times g$ for 3 h in a Beckman rotor (41 Ti) at 4 °C. The band corresponding to the virus was extracted by a syringe and concentrated by centrifuging in TNE buffer (20 mM Tris-HCl, 250 mM NaCl, 1 mM EDTA, pH 8.0) at $38,000 \times g$ for1.5 h. The resulting virus pellet was gently suspended in TNE. The purity and integrity of the viral particles were examined by negative-stain electron microscopy.

**Sample preparation and cryo-EM data acquisition.** Cryo grids were prepared using a Vitrobot Mark IV. Purified virus sample (3.5 µl) was added to the O2, Ar glow-discharged 200 mesh holey carbon grids (Quantifoil, 1.2/1.3 Jena, Germany) at 100% humidity and 4 °C. The grid was blotted for 5 s by filter paper. It was then flash frozen by plunging into liquid ethane at −174 °C. Cryo-EM data collection was performed using 300 kV Titan Krios microscope (ThermoFisher) equipped with K2 Summit camera (Gatan). The images were recorded at a magnification of ×18,000 in super-resolution mode, yielding a calibrated pixel size of 0.68 Å. Each exposure of 6.4 s was dose-fractionated into 32 movie frames leading to a total dose of 50 e-/Å². The movie frames were binned and subjected to gain reference correction, motion correction[35], anisotropic distortion correction[36]. The corrected movie frames after excluding the first 2 frames were summed with or without dose weighting to create two micrographs with a pixel size of 1.35 Å. The micrographs without dose weighting were used to determine the parameters of contrast transfer function (CTF) by program CTFFIND4[37] and the dose-weighted micrographs were used for further data processing.

**Cryo-EM data processing.** The virus particles in each micrograph were selected by e2boxer.py[38]. A total of 36,318 particles were extracted (bin 4) from 2589 micrographs in RELION[39] and subjected to reference-free 2D classification. 29974 particles were selected and subjected to 3D classification with an initial model created by starticos program[40]. All particles were selected for 3D auto-refinement in RELION[39], which led to a 10.8 Å-resolution reconstruction. The rotational, translational and CTF parameters in the last iteration of the reconstruction were extracted and adapted to JSPR format by a home-made script for unbin data. The virus particles were re-extracted without binning and the 10.8 Å-resolution map was re-scaled to a pixel size of 1.35 Å for further refinement. The rotational and translational parameters of each particle were iteratively local-refined and converged to a resolution of 4.3 Å. The block-based reconstruction method[23] was used for further refinement. Three blocks (five trimers near icosahedral five-fold axis, four trimers near icosahedral-three-fold axis and 17 CPs) were selected and refined separately to overcome the flexibility among the blocks. The final resolution of the glycoprotein shell as determined by gold standard FSC using RELION post-process is 3.5 Å[39].

**Model building and refinement.** For the glycoprotein E1, E2, and E3, crystal structures of the SINV E2-E1 heterodimer at low pH (PDB code 3MUU) and immature envelop glycoprotein complex of Chikungunya virus (PDB code 3N40) were fitted into the 3.5 Å cryo-EM map using UCSF Chimera[41]. The amino acid sequences were then mutated to their counterparts in SINV strain YN_222 in Coot. The atomic model of the rest of E1 was built manually using Coot in the Baton mode. The structures of E1, E2, and E3 were refined in real space against the cryo-EM map by Phenix with geometry and secondary structural restraints. The refinement statistics of the structural model are listed in supplementary Table 3 The structure of CPs was determined by fitting the crystal structure of conserved

core domain of Venezualan Equine Encephalitis capsid protein (PDB code 1EP5) into the capsid core.

## Data availability

The atomic coordinates of the Sindbis virus envelope glycoprotein has been deposited in the Protein Data Bank with accession code 6IMM. The cryo-EM map of the envelope glycoprotein and capsid core has been deposited in the Electron Microscopy Data Bank with accession code EMD-9693 and EMD-9692.

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

## Acknowledgements

We thank Y. Xiang and L. Li for useful discussions on this project. Cryo-EM samples were prepared at the EM lab at Harbin Veterinary Research Institute, Chinese Academy of Agricultural Sciences. Cryo-EM data collection was carried out at the Center for Biological Imaging, Core Facilities for Protein Science at the Institute of Biophysics (IBP), Chinese Academy of Sciences (CAS). We thank X. Huang, G. Ji, Z. Guo, B. Zhu, F. Sun, and other staff members at the Center for Biological Imaging (IBP, CAS) for their support in data collection. The project was funded by the National Key R&D Program of China (2017YFA0504700, 2016YFD0500705, 2017YFD0500105), the Strategic Priority Research Program of CAS (XDB08030204) and National Natural Science Foundation of China (31570874, 31460669, 31660714). X.Z. received scholarships from the 'National Thousand (Young) Talents Program' from the Office of Global Experts Recruitment in China.

## Author contributions

X.Z. and J.W. conceived the project; J.L.W., L.W., and Y.H. isolated and identified the Sindbis virus strain; Z.L. and S.W. conducted the EM analysis; M.W. and Z.S. prepared the cryo-EM samples; L.C. and J.M. performed the cryo-EM data collection; L.C. and D. Z. processed the cryo-EM data and reconstructed the cryo-EM map; J.M., L.C., and L.K. built and refined the structure model; L.C., J.M., X.Z., and J.W. analyzed the structure; X. Z. and J.W. wrote the manuscript; all authors discussed and commented on the results and the manuscript.

## Additional information

**Competing interests:** The authors declare no competing interests.

