## [Peer Review File · Nature Communications]

Reviewers' Comments:

Reviewer #1:

Remarks to the Author:

The structure of the Sindbis virus particle described here provides the highest resolution for an alphavirus particle so far, with most of the glycoprotein shell at around 3.5-3.6Å resolution. This higher resolution was achieved by using a block-based reconstruction method published earlier by the authors, which takes into account the deviations from icosahedral symmetry. Such deviations had limited the resolution of the icosahedrally averaged reconstructions published earlier. The increased resolution allows the authors to better describe the region in between the glycoprotein shell and the trans-membrane region, providing a number of detailed interactions that had not been described before. In particular, they observe interactions made by "subdomain D" of E2, near the transmembrane segment, involving a number of conserved residues that interact with E1 immediately before reaching the viral membrane. Importantly, they detect a pocket factor in this region, apparently a bound lipid molecule at the interface between E1 and E2, which is maintained in position by contacts with highly conserved residues across alphaviruses, which suggest that it is a conserved feature of this virus genus. This is a very interesting observation that will allow new structure-guided analysis of the importance of this lipid factor, and therefore opens potentially new aspects of research to identify compounds that may interfere with alphavirus assembly during exit or for disassembly during entry. The only problem that I see is the title: it is miss-leading and should be tuned down, like "implications for entry and assembly" rather than "insights" into these complex processes.

Specific issues:

There is a mistake in the legend to Figure 1, as the type I interactions are were defined in the text as being between a spike on an icosahedral axis and a spike on a quasi-3-fold axis. The legend reads like it was between two quasi-3-fold spikes, which is the definition of type II.

Figure 1C appears to confirms results from fitting the x-ray structure of SFV E1 into a low resolution cryo-EM map of SFV, which showed E1 H125 from two adjacent spikes facing each other and suggesting a way of disassembly by low pH (Roussel et al, 2006). This work should at list be cited. The authors mention a "hydrophobic pore in NP" without really defining it, but apparently meaning the pocket that accommodates the C-terminal end of E2 in the capsid. This should be better explained, as the term "pore" is confusing (a "pore" should allow something to pass through, which is not the case here)

Reviewer #2:

Remarks to the Author:

The manuscript "Structural insights into alphavirus host-cell entry and assembly revealed by a 3.5 Å resolution cryo-EM structure" by Chen, et al., describes a cryo-EM reconstruction of Sindbis virus (SINV), with an resolution of 3.5 Å. This is an improvement from previously published highest resolution of 7 Å for SINV, and, for an alphavirus in general, from 4.8 Å for Venezuelan equine encephalitis virus. Although there is a significant improvement in resolution, as all alphavirus structures are extremely similar, much of the findings presented here are not new. The contribution to the knowledge about the Sindbis structure or alphavirus in general, is therefore only incremental. The authors described the structure following the established terminology in alphavirus structural biology, and highlighted several new findings based on some atomic interactions observed. Alphaviruses have icosahedral nucleocapsid covered by a lipid membrane, surrounded by a glycoprotein shell. Alphavirus shell is spherical with 80 spikes protruding from it. It is composed from 240 copies of heterodimers of proteins E1 and E2 and follows Caspar-Klug triangulation rule with T=4, meaning here that there are four E1-E2 heterodimers in an icosahedral asymmetric unit. In the structure, E1 lies flat on the virus surface and E2 protrudes outwards. Both proteins are anchored in

the lipid membrane by transmembrane helices that reach to and interact with the nucleocapsid. Three of the heterodimers form a triangle with a trimeric spike in the center, and the fourth heterodimer is in icosahedral three-fold triangle and spike, the two subunits related by a pseudo-2-fold symmetry. Compared to lower resolution structures, the interactions between trimeric subunits within and between icosahedral asymmetric units are much clearer and described in greater details. It should be noted that the authors call the whole triangular subunit of E1 with the E2 spike in the center, a spike. That results in a misleading "inter-spike interactions" described in the manuscript.

The authors also described a hydrophobic pocket between E1 and E2 in the heterodimer, composed from conserved amino acid residues, to form interactions with a piece of phospholipid which they named "pocket factor". That region of the E1/E2 structure consisted of membrane associated stem regions and transmembrane regions and therefore, they are supposed to interact with the viral membrane, and so the significance of this special factor is probably over-sold. They also assign it a role in virus stability and infection for which experiment evidence is completely missing.

This reviewer feels that the English in this manuscript needs significant improvement and will greatly benefit by asking a native English speaker for help in editing. There are multiple mistakes in the choice of words to use; there are sentences that are not clear in meaning, and there is a lot of repetition.

Overall, the quality of the structure presented is good, and the improvement in resolution is significant. However, the findings obtained from this structure are only incremental and therefore the manuscript is more suited in a specialized journal.

Reviewer #1 (Remarks to the Author):

The structure of the Sindbis virus particle described here provides the highest resolution for an alphavirus particle so far, with most of the glycoprotein shell at around 3.5-3.6Å resolution. This higher resolution was achieved by using a block-based reconstruction method published earlier by the authors, which takes into account the deviations from icosahedral symmetry. Such deviations had limited the resolution of the icosahedrally averaged reconstructions published earlier. The increased resolution allows the authors to better describe the region in between the glycoprotein shell and the trans-membrane region, providing a number of detailed interactions that had not been described before. In particular, they observe interactions made by “subdomain D” of E2, near the transmembrane segment, involving a number of conserved residues that interact with E1 immediately before reaching the viral membrane. Importantly, they detect a pocket factor in this region, apparently a bound lipid molecule at the interface between E1 and E2, which is maintained in position by contacts with highly conserved residues across alphaviruses, which suggest that it is a conserved feature of this virus genus. This is a very interesting observation that will allow new structure-guided analysis of the importance of this lipid factor, and therefore opens potentially new aspects of research to identify compounds that may interfere with alphavirus assembly during exit or for disassembly during entry. The only problem that I see is the title: it is miss-leading and should be tuned down, like “implications for entry and assembly” rather than “insights” into these complex processes.

Response: We thank the reviewer for the positive comments. We have now changed the title of the revised version to “Implication for alphavirus host-cell entry and assembly indicated by a 3.5 Å resolution cryo-EM structure“.

Specific issues:

There is a mistake in the legend to Figure 1, as the type I interactions are were defined

in the text as being between a spike on an icosahedral axis and a spike on a quasi-3-fold axis. The legend reads like it was between two quasi-3-fold spikes, which is the definition of type II.

Response: We thank the reviewer for pointing out this mistake. We have now changed the legend in the revised manuscript.

Figure 1C appears to confirm results from fitting the x-ray structure of SFV E1 into a low resolution cryo-EM map of SFV, which showed E1 H125 from two adjacent spikes facing each other and suggesting a way of disassembly by low pH (Roussel et al, 2006). This work should at list be cited.

Response: We apologize for this oversight, and agree this reference should be included. The reference has been added in the revised manuscript.

The authors mention a “hydrophobic pore in NP” without really defining it, but apparently meaning the pocket that accommodates the C-terminal end of E2 in the capsid. This should be better explained, as the term “pore” is confusing (a “pore” should allow something to pass through, which is not the case here)

Response: We thank the reviewer for the suggestion. We have now changed the term ‘pore’ was to “pocket”.

Reviewer #2 (Remarks to the Author):

The manuscript “Structural insights into alphavirus host-cell entry and assembly revealed by a 3.5 Å resolution cryo-EM structure” by Chen, et al., describes a cryo-EM reconstruction of Sindbis virus (SINV), with an resolution of 3.5 Å. This is an improvement from previously published highest resolution of 7 Å for SINV, and,

for an alphavirus in general, from 4.8 Å for Venezuelan equine encephalitis virus. Although there is a significant improvement in resolution, as all alphavirus structures are extremely similar, much of the findings presented here are not new. The contribution to the knowledge about the Sindbis structure or alphavirus in general, is therefore only incremental.

Response: Thank for your comment that our work is a significant improvement in resolution.

As you said that all alphavirus structures are extremely similar which indicates that the detailed structure features we found on this virus are very possibly common for all alphaviruses.

One of the most important part of the structural biology is identifying the key functional related interactions of proteins. An atomic model is required for accurately identifying an amino acid-amino acid interaction. Our work pushed the resolution of the reconstruction of an alphavirus to 3.5 Å, which enable us to build an atomic model of the whole alphavirus virus accurately. With this atomic model, we can determine several key interactions which have never been found or conformed. Most of the interactions are formed by completely or highly conserved amino acids. Some of the interactions have been already proposed based on the low resolution result and have been proven important for the virus life cycle by functional studies.

The authors described the structure following the established terminology in alphavirus structural biology, and highlighted several new findings based on some atomic interactions observed. Alphaviruses have icosahedral nucleocapsid covered by a lipid membrane, surrounded by a glycoprotein shell. Alphavirus shell is spherical with 80 spikes protruding from it. It is composed from 240 copies of heterodimers of proteins E1 and E2 and follows Caspar-Klug triangulation rule with T=4, meaning here that there are four E1-E2 heterodimers in an icosahedral

asymmetric unit. In the structure, E1 lies flat on the virus surface and E2 protrudes outwards. Both proteins are anchored in the lipid membrane by transmembrane helices that reach to and interact with the nucleocapsid. Three of the heterodimers form a triangle with a trimeric spike in the center, and the fourth heterodimer is in icosahedral three-fold triangle and spike, the two subunits related by a pseudo-2-fold symmetry. Compared to lower resolution structures, the interactions between trimeric subunits within and between icosahedral asymmetric units are much clearer and described in greater details. It should be noted that the authors call the whole triangular subunit of E1 with the E2 spike in the center, a spike. That results in a misleading “inter-spike interactions” described in the manuscript.

Response: We have now changed “inter-spike interactions” to “inter-triangle interactions” in the revised manuscript.

The authors also described a hydrophobic pocket between E1 and E2 in the heterodimer, composed from conserved amino acid residues, to form interactions with a piece of phospholipid which they named “pocket factor”. That region of the E1/E2 structure consisted of membrane associated stem regions and transmembrane regions and therefore, they are supposed to interact with the viral membrane, and so the significance of this special factor is probably over-sold. They also assign it a role in virus stability and infection for which experiment evidence is completely missing.

Response: Based on our structure, we can see that the pocket factor forms several hydrophobic interaction with the components that form the hydrophobic pocket, thus stabilizing this pocket. The amino acids that are involved in these interactions are either completely or highly conserved among alphaviruses. This stability is important for both virus assembly and virus entry. The reasons are listed as below.

A previous study of the cryo-EM structure of SFV at pH of 5.9 showed a

conformational change of the transmembrane region that forms this pocket (Haag et al., 2002). This finding indicates the interaction between the pocket factor and the pocket has to be interrupted during virus entry. We added this new evidence into the manuscript.

On the other hand, the stability of the pocket is important for virus assembly. The pocket is formed by E2 D subdomain, E1 TM and E1 Stem loop. Interactions between E2 P317 and E1 W409, and between E2 H314 and E1 S403 are completely conserved. A previous study conducted on SFV showed that the mutation of E2 H348 (corresponding SINV E1 H314) to alanine causes an unexpected defect in the budding process. Such a mutation disrupts the interaction between E2 D-loop and E1 thus destabilizing the pocket.

Together, these studies clearly demonstrated that the stability of the hydrophobic pocket is important for both virus budding and virus entry. Our structure shows that the pocket factor plays important role in stabilizing this pocket and therefore may be essential for both virus budding and virus entry.

This reviewer feels that the English in this manuscript needs significant improvement and will greatly benefit by asking a native English speaker for help in editing. There are multiple mistakes in the choice of words to use; there are sentences that are not clear in meaning, and there is a lot of repetition.

Response: We thank the reviewer for this suggestion. The manuscript has now been carefully edited by a native English speaker, hopefully to good effect.

Overall, the quality of the structure presented is good, and the improvement in resolution is significant. However, the findings obtained from this structure are only incremental and therefore the manuscript is more suited in a specialized journal.